# Exogenous Melatonin Use in University Students: A Cross-Sectional Survey

**DOI:** 10.3390/pharmacy12020041

**Published:** 2024-02-23

**Authors:** Sulafa T. Alqutub, Faris A. Alzahrani, Abdulrahman S. Hassan, Abdullah H. Alirbidi, Osama A. Alraddadi, Omar A. AlSadah, Mohammad B. Yamani, Mansour Tobaiqy

**Affiliations:** 1Department of Family and Community Medicine, College of Medicine, University of Jeddah, Jeddah P.O. Box 45311, Saudi Arabia; 2College of Medicine, University of Jeddah, Jeddah P.O. Box 45311, Saudi Arabia; 2040418@uj.edu.sa (F.A.A.); 2040060@uj.edu.sa (A.S.H.); 2040267@uj.edu.sa (A.H.A.); 2040837@uj.edu.sa (O.A.A.); 2041929@uj.edu.sa (O.A.A.); 2040067@uj.edu.sa (M.B.Y.); 3Department of Pharmacology, College of Medicine, University of Jeddah, Jeddah P.O. Box 45311, Saudi Arabia; mtobaiqy@uj.edu.sa

**Keywords:** exogenous melatonin, university students, Saudi Arabia, sleep quality

## Abstract

To assess the prevalence of melatonin use and its perceived benefits among university students in different specialties in Saudi Arabia, a cross-sectional survey was conducted between March and June 2023. Data about demographics, time of melatonin use, perceived reasons for exogenous melatonin use, melatonin use in relation to exam periods, perceived safety, and adverse effects was gathered. Of 380 students, ~52% reported using exogenous melatonin for sleep disorders. Most participants reported using melatonin during and after exam periods. Additionally, several (*n* = 157; 75.4%) believed that its use was safe. The predominant use patterns were daily and as needed, and this study observed a higher rate of use compared with previous studies in Saudi Arabia. The most frequently reported adverse effect was headache (*n* = 36; 37.5%). A significant number of undergraduate male students in health specialties used melatonin. A high rate of melatonin use was noted during exam periods, which was attributed to sleep deprivation. Additionally, a significant number of students from private universities reported using exogenous melatonin. Melatonin use is common among university students. Future research should use a reliable psychometric measure to test its effect on university students’ sleep quality and quantity.

## 1. Introduction

Insomnia is a common sleep disorder affecting 10–17.6% of the adult population, defined as difficulty falling asleep and/or staying asleep [1]. Sleep quality is critical for university students’ health, as poor sleep quality negatively impacts cognitive and academic performance [2,3]. In Saudi Arabia (SA), Alsaggaf and Wali reported that most university medical students are adversely impacted by at least one form of sleep disorder [4]. Excessive daytime sleep was the most frequently reported disorder at 40%, followed by insomnia at 33% and poor sleep quality at 30% [4]. Another study, conducted at the University of Umm Al-Qura in SA, reported that 22.4% of participating students suffered from circadian rhythm sleep disorder [5]. In Qatar, Ali et al. reported a 70% prevalence of poor sleep quality and inadequate sleep hygiene practices among university-aged students. Their study also identified a significant correlation between the use of herbal medicine as a sleeping aid and high scores on the Pittsburgh Sleep Quality Index (PSQI) [6].

The use of over-the-counter (OTC) self-medication and/or prescription medication is relatively frequent behavior among university students within SA. The use of self-medication and/or prescription medication is defined as “the selection and use of medicines/medicinal products, including herbal and traditional products by individuals, to treat self-recognized illness or symptoms, or the intermittent or continued use of a medication prescribed by a physician for chronic or recurring diseases or symptoms” [7]. In 2022, Alasmari et al. reported a prevalence of self-medicating with sleeping pills among university students of about 86%, where more than 89% of study participants misused the pills. Relative to other sleeping pills, melatonin was, after Panadol, the second most prevalent sleeping pill used at 74% and 58%, respectively [8]. A recently published study showed that headache was the most frequent indication for self-medication at 35.6%, especially during examination periods. The most commonly used medications were analgesics, antipyretics, cough and cold medications, and vitamins [9].

From its early use for sleep disorders in the early 1980s, exogenous melatonin has been an OTC supplement, and is identified as relatively non-toxic with mild side-effects [10]. It is widely used in the management of insomnia, jet lag, and some other forms of secondary sleep disorders including obstructive sleep apnea (OSA). The typical available doses used in various studies have a range of 0.1–10 mg [1]. Conversely, a randomized controlled trial (RCT) with a crossover design highlighted conflicting evidence regarding the effectiveness of melatonin in promoting sleep. The study indicated that melatonin was not effective in sustaining sleep or improving self-reported night-time sleep quality [11]. In addition to its chronobiotic and central nervous system effects, melatonin has secondary effects on the cardiovascular and gastrointestinal systems, potentially leading to adverse effects (AEs) [12].

More recent evidence indicates the safety and effectiveness of exogenous melatonin as a pharmacological supplement in regulating the circadian rhythm of the sleep–wake cycle in adults and children older than 11 years [13,14,15]. Moreover, melatonin showed positive impacts on sleep quality for other common forms of sleep disorders, such as insomnia, breathing-related sleep disorders, and sleep-related movement disorders [16,17,18]. Moreover, evidence from 26 RCTs concluded that exogenous AEs can be avoided when it can be administered in accordance with the biological circadian rhythm [19].

Notably, melatonin supplements have been incorporated into recent operational definitions of complementary, alternative, and integrative medicine (CAIM), as cited by Cochrane Complementary Medicine, to improve the quality of reporting in systematic review studies [20,21]. Additionally, in 2022, the Saudi Food and Drug Authority (SFDA) classified melatonin of 2 mg or less as a food supplement [22].

Given melatonin’s primary chronobiotic effect and its secondary impacts on cardiovascular and gastrointestinal functions, future studies are expected to provide further evidence regarding its safety and efficacy [1,12]. These studies may explore its efficacy and safety in specific medical conditions, such as its potential use as an adjuvant therapy in SARS-CoV-2, neuropsychiatric diseases, and cancer, as well as its role in improving patient outcomes among patients with obesity and diabetes and reducing pre- and postoperative anxiety in adults [23,24,25,26,27,28].

There is not much known about university students and their use of melatonin; hence, the main aim of this study is to assess the pattern of melatonin use, the reliance on it as a sleeping aid, and the perceived benefits and the adverse effects in the management of sleep disorders among students in different specialties within Saudi Arabia. 

## 2. Materials and Methods

### 2.1. Study Site and Design

A cross-sectional survey and face-to-face interviews were conducted in university students studying at both governmental and private universities within Mecca and Jeddah cities, SA, from March to June 2023. For logistical and budgetary reasons, the two approximate cities were selected. Both cities are located close to the west coast of SA. According to the Jeddah Municipality directory webpage, 14 governmental and private universities are located in Jeddah [29], whereas Mecca has only one university. 

The Equator Guidelines for Strengthening the Reporting of Observational Studies (STROBE) were applied while preparing this manuscript for submission [30].

### 2.2. Sample Size 

A report conducted by the SA Ministry of Education showed that there are currently about 217,657 university students studying for their bachelor’s degrees, forming the target population. The Open Epi info main menu was used to calculate the sample size for the proportion. Using the previous estimate of melatonin use among university students of 58% [8], the following formula was used to calculate the sample size:n = [DEFF × Np(1 − p)]/[(d^2^/Z^2^_1−α/2_ × (N − 1) + p × (1 − p)] 

Considering the 95% confidence level at a 5% margin of error, the calculated sample size was 374. A total of 462 students were approached to compensate for the possibility of missing information.

### 2.3. Questionnaire Development 

The questionnaire was developed in Arabic as it is the native language in Saudi Arabia. It included both items requiring responses on a 5-point Likert scale and closed-ended questions with multiple lists of items. The first section of the questionnaire gathered information on the sociodemographic characteristics of the students (sex, nationality, marital status, working part-time, undergraduate specialty, name of the university). The second section included 10 questions on patterns of exogenous melatonin use from the students’ perspective, use frequency, the main reasons for exogenous melatonin use, and the use of melatonin in relation to examination periods. Certain statements were added to uncover potential melatonin dependence, perceived safety, and whether respondents suggested melatonin to other college students. Finally, the list of potential adverse effects included headache, dizziness, sleepiness, nightmares, mood swings, constipation, loss of appetite, and gastrointestinal upset. The questions further investigated the intensity of adverse effects (mild, moderate, and severe). Before administration, the questionnaire was reviewed for face and content validity by a panel of 10 individuals from the medical center at the University of Jeddah, representing academic, healthcare, and administrative staff. The questionnaire was then piloted with 10 university students. The instrument’s reliability was evaluated using Cronbach’s alpha coefficient for items rated on a 5-point Likert scale. A calculated reliability score of 0.71 was deemed acceptable. The pilot responses were included in the analysis dataset, as no changes were made to the questionnaire post-piloting. The questionnaire’s translation into English was only for publication purposes.

### 2.4. Recruitment 

Convenience sampling was used to contact eligible participating university students (*n* = 462), and face-to-face interviews were facilitated by the research team members on weekdays. Undergraduate university students were targeted; those who were unwilling to participate were excluded. The interviews were conducted, and the responses were captured electronically by the research team members. All participants provided informed consent for their participation, which was voluntary.

### 2.5. Analysis

Descriptive statistics with frequency tables were developed for various sociodemographic variables. Chi-square testing was used to test for a significant association between the outcome variable, namely, the reported exogenous regular daily use of melatonin in the management of sleep disorders, and the other independent variables, such as gender, nationality, working status, specialties, and university of study. *p*-values < 0.05 were considered to indicate significance. Data analysis was undertaken using SPSS (SPSS Inc., Cary, NC version 22.0) [Computer software].

## 3. Results

In all, 462 students were approached to take part in the study, but only 380 agreed to participate, forming a response rate of 82.3% (380/462). Most participants (*n* = 138; 66.3%) reported using melatonin as needed (PRN), some (*n* = 45; 21.6%) reported using it once a day, and few (*n* = 25; 12%) reported using it either twice or thrice a day (Table 1).

### 3.1. Demographics of the Students 

Table 1 shows the demographic characteristics of the participating university students. Most participants were male (64.5%) and Saudi nationals (94.5%). The reported prevalence of melatonin use was 51.6%, while 17.9% reported using hypnotics for sleep disorders. Among the 196 regular users, the primary perceived reasons for exogenous melatonin use were as follows: 59.7% used it to achieve better sleep quality, 33.2% for insomnia, and 7.1% reported habitual use as the main reason.

### 3.2. Duration and Recommendation of Melatonin Use by Students

Table 2 presents participants’ attitudes and behaviors toward melatonin’s safety and effectiveness. Most participants reported using melatonin during examination periods (62.5%), and after examinations (72.8%). More than four-fifths of the participants (81.8%) reported that melatonin improved their sleep quality. Additionally, 75.4% of them believed that melatonin was safe to use, and 24.5% reported misuse of it. About 71% of participants recommended melatonin to other university students having difficulty sleeping, and only 36.6% relied on melatonin to sleep. 

### 3.3. Adverse Effects and Their Intensities Reported by Students 

Out of 196 melatonin users, 97 (49.4%) reported AEs, as shown in Table 2. The most frequently reported AE by this latter group was headache (*n* = 36;3 7.5%), followed by dizziness (*n* = 20; 20.8%), sleepiness (*n* = 17; 17.7%), nightmares (*n* = 10; 10.4%), loss of appetite (*n* = 6; 6.3%), and other GIT symptoms (*n* = 7; 7.3%). Most reported adverse effects were mild 64 (66%), while 33 (34%) were moderate, and none were severe.

### 3.4. Factors Associated with the Use of Exogenous Melatonin

Following the application of chi-square testing, the association between various predictors and the main outcome variable, namely, regular melatonin use, was assessed. Undergraduate male students studying in medical and/or health specialties were more likely to report using exogenous melatonin, and the differences within gender and between specialties were statistically significant at *p* < 0.007. Relative to other universities, students at King Abdulaziz University (KAAU) and private universities were more likely to use exogenous melatonin, with a statistically significant difference at *p*-value < 0.001. 

## 4. Discussion

These results provide valuable insights into the prevalence, patterns, and misuse of, and attitudes toward, exogenous melatonin among university students. More than half of the participating students reported self-medication with melatonin to manage their sleep disorders. This finding is consistent with previous studies of melatonin use among university students, where self-medication with melatonin has been reported as 58% [8]. At least 22% of the university students in the sample of this study regularly use melatonin as a sleeping aid on a daily basis. By contrast, its regular use is significantly greater than previously observed patterns of 7% [8]. The observed higher rates of use could be attributed to misuse, perceived safety, observed low levels of adverse effects, and easy access. Further investigation is needed to explore the reasons for the increase in the pattern of regular use. In addition, further studies should be conducted among medical students and students at KAAU and private universities to explore the leading factors associated with the excess use of exogenous melatonin.

The significant high exogenous melatonin use by male students is inconsistent with previous studies showing that female students are more likely to consume larger doses of sleeping pills [8]. The difference between these studies can be explained by gender differences in mental health status and some medical conditions, such as anxiety, depression, and OSA, as well as their impact on sleep quality. Hence, the demand for sleep-support agents may increase [31]. Another significant finding reported the high use of exogenous melatonin among students studying in health and/or medical schools compared with students studying in non-health-related specialties. Similar studies have shown an increase in the self-prescription of sleeping pills as well as stimulants among medical students to cope with the highly demanding learning environment and to meet the expectation of managing the academic load. Medical students appear to consume a high quantity of sleeping aids, possibly owing to their greater familiarity with hypnotic agents by virtue of their studies [8,9,32].

Ironically, the reported prevalence of hypnotic agent and/or sedative use in this study mirrored findings from the first study of this kind, which was conducted 12 years ago among medical students in the Riyadh region, SA [33]. This continued pattern could be attributed to the availability and tolerability of exogenous melatonin, which replaced demand for other sleep aid agents [14].

A cross-sectional study examined insomnia prevalence and treatment practices among 2029 Portuguese higher education students; 31% of the students reported experiencing insomnia symptoms, and only 6% of them used physician-prescribed sleep medication, primarily benzodiazepines. General practitioners were the most frequent prescribers, and 4% of the students consumed OTC sleep aids or supplements. Valerian was the most common herbal remedy, and melatonin was also reported [34].

Misuse is defined as “the use of a substance for a purpose not consistent with legal or medical guidelines” [7]. Regarding previously identified indicators of misuse, they are the daily use of sleeping pills, their use in greater amounts and exceeding the recommended dose, self-medication, and their use for recreational purposes. Unfortunately, we were only able to capture information concerning the prevalence of regular daily use and self-prescription. Thus, the reported 24.5% of perceived misuse could be an underestimation.

Habitual use of melatonin was reported by 7% of the participating regular melatonin users. A previous double-blind placebo-controlled study concluded that consuming a small physiological concentration of oral exogenous melatonin at habitual bedtime could effectively reduce the time for sleep latency and latency to stage 2 sleep [35].

This study also found that melatonin is perceived as safe and effective by university students. About 60–70% of these students reported agreeing or strongly agreeing concerning its effectiveness in improving the quality of sleep during examination periods and afterward. The most likely driving factor is sleep deprivation associated with the examination period.

Notably, the results from this double-blind randomized controlled study concluded that 6 mg melatonin intake is effective in improving the psychomotor and physical performance of sleep-deprived athletic students [36].

It is important to note that adverse effects were reported by less than half of the participating group, with the most common being of mild intensity. This is consistent with Besag et al.’s findings drawn from 37 RCTs, where the reported short-term adverse effects from exogenous melatonin use were mild to moderate [37]. The perceived safety and effectiveness were also reflected, where more than 60% of the study group reported recommending the use of exogenous melatonin to other students, reporting strongly agreeing or agreeing with the statement that they recommended melatonin. This finding is aligned with the findings of a systematic review on self-medication misuse in the Middle Eastern region that identified friends and parents as secondary sources of drug information [38]. 

As reported by the participating students, headache was the most frequent adverse event associated with exogenous melatonin use. A previous review showed a 40% increase in AEs in control trials using 10 mg or higher doses of melatonin supplements [37]. Our finding is similar to the most frequently reported adverse event in a systematic review and meta-analysis on the safety of higher doses of melatonin [39]. This finding suggests that participating students are consuming higher doses of melatonin, or they are using low doses frequently; hence, more AEs are reported.

The most commonly reported pattern of use among regular users is “as needed” (*n* = 138; 66.3%), which supports the essential role of health professionals, namely, community pharmacists, in increasing consumers’ awareness about the misuse, overuse, and drug interaction of melatonin and other over-the-counter medications [40].

This study has certain limitations, including the use of a cross-sectional design, which made it difficult to identify the short-term and long-term benefits and the potential AEs of exogenous melatonin among university students. Data regarding students’ years of study, history of chronic diseases, melatonin dose, and drug formulary were not considered. Thus, the effect of melatonin intake on the secondary causes of sleep disorders, such as OSA, could not be described. With the recently identified operational definition of CAIM for melatonin, it was difficult to find previous systematic reviews and evidence on the standard list of therapies using exogenous melatonin. This has resulted in the inconsistent reporting of melatonin use in the literature, with some studies considering sleeping pills to be exogenous melatonin and others considering food supplements or herbal medicines to be in this category [6,8,41]. Despite these limitations, this study sheds light on the prevalence of regular melatonin use among university students in SA; this may inform future research and lifestyle medicine interventions targeting university students.

Future studies could adopt a longitudinal design using reliable psychometric measures of sleep quality and quantity to better understand the impact of melatonin use on sleep and academic performance. Assessments of sleep quality may include proper assessments of sleep latency, sleep efficiency, sleep waking frequency, hours of sleep per night, and wakefulness time, or the use of more structured tools, such as the validated PSQI, to measure the effect of exogenous melatonin on the quality of sleep [6,42]. 

A review that evaluated the efficacy of melatonin and melatonin agonists in primary and secondary insomnia found that melatonin shows statistically significant improvements in sleep latency and total sleep time. However, there is no consensus on whether these improvements are clinically meaningful. Similar results were observed for ramelteon, a melatonin agonist. This review concluded that the existing evidence is limited due to varying methodological quality and a need for an improved consensus on outcome measures for insomnia. Despite these limitations, the short-term use of melatonin was generally associated with improved sleep quality and latency, but reports regarding its effect on total sleep time are varied [43].

Moreover, given the significant person-to-person variability in melatonin bioavailability and its pharmacokinetic nature, differences across extreme age groups are anticipated. This variability is linked to differences in hepatic metabolization, which, in turn, correlates with hepatic function variability [44]. Hence, studies in children and elderly populations are crucial. Future research may uncover the pattern and indication for exogenous melatonin use in the management of anxiety and other neuropsychiatric diseases, cancer, and obesity with diabetes.

Within universities and colleges, it is essential to offer students information, programs, and resources on sleep hygiene and the appropriate use of sleeping aids. Moreover, to ensure the validity of data collection, future research should consider applying the standard operational definition of CAIM for melatonin [21].

The rising prevalence of sleep disturbance and melatonin use among university students warrants further investigation and intervention strategies. Future research should explore the factors influencing melatonin use, including academic stress and underlying mental health conditions, in depth. Additionally, studies should assess the long-term effects of melatonin use in this population, particularly with regard to potential dependence on or interactions with other medications.

Universities and health services can play a pivotal role in addressing this issue. Proactive measures include educational programs such as workshops and seminars on healthy sleep habits, the risks and benefits of melatonin, and alternative sleep-promoting strategies. These initiatives can empower students to make informed decisions about their sleep health. Increasing access to mental health services and promoting help-seeking behaviors can address the underlying conditions contributing to sleep disturbance and consequent melatonin use. Additionally, online platforms offering cognitive behavioral therapy for insomnia or mindfulness-based interventions can provide accessible and practical support for students experiencing sleep difficulties. The effectiveness and necessity of specific intervention programs will depend on further research and careful consideration of local contexts and student needs. Notably, a multifaceted approach that combines education, mental health support, and accessible therapeutic interventions can help improve sleep health and reduce the reliance on melatonin among university students [45].

## 5. Conclusions

More than half of the university students in this study reported that the main reason for their use of exogenous melatonin was to improve sleep quality. Male students in health specialties were significantly more likely to report using exogenous melatonin. The examination period showed relatively higher rates of use, which could be attributed to sleep deprivation. The most frequent patterns of use were daily and as needed, with a higher rate of use observed compared with previous studies among students in the country. Further research is required using a reliable psychometric measure to test melatonin use’s impact on university students’ sleep quality and quantity. 

## Figures and Tables

**Table 1 pharmacy-12-00041-t001:** Participants’ demographic characteristics (*n* = 380).

Variable	Frequency (%)
Sex	Male 245 (64.5) Female 135 (35.5)
Nationality	Saudi 359 (94.5) Non-Saudi 21 (5.5)
Marital status	Married 4 (1.1) Unmarried 376 (98.9)
Working status	Part-time worker 38 (10) Not working 342 (90)
Undergraduate specialty	Health and/or medical school 172 (45.3) Non-health school 208 (54.7)
Name of the university	University of Jeddah (UOJ) 146 (38.4) King Abdulaziz university (KAAU) 86 (22.6) Umm AlQura 56 (14.7) King Saud National Guard 23 (6.1) Private 56 (14.7) Other governmental university 13 (3.4)
Reported regular daily exogenous melatonin use in the management of sleep disorders	Yes 196 (51.6) No 184 (48.4)
Reported daily hypnotic and/or sedative use in the management of sleep disorders	Yes 68 (17.9) No 312 (82.1)
Patter of use among daily users (timing)	As needed 138 (66.3%) Once per day 42 (21.6%) Twice per day 23 (11.1%) Three times per day 2 (1%)

**Table 2 pharmacy-12-00041-t002:** Attitude to the safety and effectiveness of exogenous melatonin use in the study group (*n* = 208).

Item	Strongly Agree	Agree	Neutral	Disagree	Strongly Disagree
(1) I use melatonin during exam period	25 (12%)	105 (50.5%)	16 (7.7%)	43 (20.7%)	19 (9.1%)
(2) I use melatonin after exams	25 (12.1%)	125 (60.7%)	22 (9.7%)	24 (11.7%)	12 (5.8%)
(3) Melatonin has improved my sleep quality	54 (26%)	116 (55.8%)	24 (11.5%)	8 (3.8%)	6 (2.9%)
(4) Melatonin is safe to use	50 (24%)	107 (51.4%)	37 (17.8%)	11 (5.3%)	3 (1.4%)
(5) Have you ever misused melatonin	3 (1.4%)	48 (23.1%)	52 (25%)	52 (25%)	53 (25.5%)
(6) I would recommend melatonin to other university students who have sleeping difficulties	49 (23.6%)	98 (47.1%)	48 (23.1%)	7 (3.4%)	6 (2.9%)
(7) I rely on melatonin to sleep	18 (8.7%)	58 (27.9%)	24 (11.5%)	62 (29.8%)	46 (22.1%)

## Data Availability

Responses collected from the participants are available upon request. Questionnaire items are also available.

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
