# Peer review of "Exogenous Melatonin Use in University Students: A Cross-Sectional Survey"

_pharmacy, 2024, doi:10.3390/pharmacy12020041_

Round 1

Reviewer 1 Report

Comments and Suggestions for Authors

This paper reports the patterns of use of melatonin by students attending university in two cities of Saudi Arabia. The authors used face-to-face interviews with a de novo questionnaire of their design to assess demographic information and melatonin usage. The questionnaire seems to satisfy face validity. However, there is no report about inter-rater reliability of the scale since presumably all 380 students were not interviewed by one rater. This is pertinent particularly to the question about misuse of melatonin (see below).  About half the students surveyed have used melatonin, primarily as a hypnotic agent and believe it to be both safe and effective. With respect to the questionnaire one item asks: "Have you ever misused melatonin?" without any further explanation. In the discussion the authors note that this means use in a manner not approved by medical or legal guidelines. However, this needs some further elaboration since in Saudi melatonin is (according to the authors) classified as a food supplement. What were the defining parameters used by the interviewers to decide if students had misused melatonin?    

The authors note that melatonin use is assessed from the student perspective, but they have not reported if the substance may have been used for other reasons (e.g., as an anxiolytic; for antioxidant properties).

The introductory paragraph does seem to represent information from reference 6 correctly. Reading this paper the authors had relatively few subjects who used 'hypnotic' agents and so the correlation as such is probably biased. Furthermore, the paper is flawed in that the authors used an inappropriate correlation. Furthermore, in the current paper the statement line 49ff does not accord with the previously quoted study (ref 6) where about 80% of students reported no use of 'sleep aids' in the past three months! This does not seem to represent a frequent use of OTC or prescription meds.

The paragraphs, commencing line 83 through to line 93, do not appear to be relevant to the current topic and could be omitted. 

Line 102: the rationale for including the approximate geographic location of the study cities is not apparent.

Line 211: Reference to "sleeping pills" in ref 8 may not apply specifically to melatonin but to melatonin and other OTC and prescribed hypnotic agents e.g., benzodiazepines or Z drugs.

Line 217: Another reason that medical students appear to be greater consumers of sleeping aids might be their greater familiarity with hypnotic agents by virtue of their studies.

Line 231: The relationship between the Zhdanova et. al., study and the finding of habitual use of melatonin in 7% of respondents seems tenuous. Yes, the study provides a rationale for the use of a low dose of melatonin 2 to 3 hrs before bedtime, but the connection to habitual use? Does habitual use imply a dependence / withdrawal syndrome with melatonin? The available clinical data would suggest no dependence issues with melatonin. How then might habitual use be explained at a physiological or pharmacokinetic level or at this level at all? Were the students questioned about potential withdrawal symptoms on abruptly stopping melatonin?

Line 246 ff: this sentence could be more succinctly expressed e.g., More than 60% of students agreed or strongly agreed to recommending melatonin to other students, attesting to the perceived safety and efficacy of the compound.   

Paragraph commencing line 282 does not seem to be relevant to the current topic. While the points raised may be valid this was not the subject of the current investigation. 

Some minor details:

Presumably 'Macca' line 101 should be 'Mecca'?

Line 145 rather the negative one could write this in a more positive way: "Participation was voluntary".

Line 258: what do you mean by 'more frequent bases'? Is this meant to be 'on a more frequent basis"?

Reference 6 does not record the authors of the study.

Reference 16 seems to be incomplete.

Comments on the Quality of English Language

The English is generally adequate but there are some instances of incorrect grammatical usage. There are numerous English spelling errors throughout the manuscript. I would suggest that composition of the manuscript on Word should detect most of these errors when setting the language option to either English US or English UK.

Reviewer 2 Report

Comments and Suggestions for Authors

- Overall, I think this is an interesting preliminary study. However, the current writing jeopardizes the comprehension of it. Here are my suggestions and comments for the authors:

1. The English should be improved. Sometimes, it is difficult to follow the text. Besides, there are several typos throughout the manuscript. Please revise the text carefully.

2. The title of the manuscript is somehow confused. Please change it. Suggestion: “Exogenous Melatonin Use in University Students: A Cross-Sectional Survey”.

3. I think that the findings from these published papers:

https://www.sciencedirect.com/science/article/pii/S2667343621000123

https://pubmed.ncbi.nlm.nih.gov/31715492/

may be important to the current study introduction and/or discussion.

4. I think that a reference to AASM guidelines and/or European Guidelines for insomnia treatment would be an important add. What do the guidelines recommend about the use of melatonin in insomnia? This should be explicit in the text (in my viewpoint).

5. The authors refer that they used inferential statistics such as Chi-Square tests, but in the Results section there are no references to these calculations. Please correct or justify the reasons.

6. It is important to discuss more deeper the implications of the findings: The prevalence of melatonin use is high, so what can universities and health services may do? What type of intervention programs could be developed? Would they be useful and/or necessary? Please add references as well.

7. Please take a look at works by Daniel Taylor (USA) and so forth. They have published several papers on sleep and insomnia in university students. Maybe some of these works may be useful for your manuscript purposes.

8. It would also be interesting to discuss whether there are significant differences in melatonin usage in countries such as Saudi Arabia and other countries (ex: European countries or USA). What does the literature say about that? In positive case, what reasons may be pinpointed?

Comments on the Quality of English Language

The manuscript has several typos and sometimes the sentences are difficult to follow. Please revise carefully the whole manuscript.

Round 2

Reviewer 2 Report

Comments and Suggestions for Authors

Overall, the authors responded to all my concerns.